# Preparation and Characterization of Soluble Dietary Fiber Edible Packaging Films Reinforced by Nanocellulose from Navel Orange Peel Pomace

**DOI:** 10.3390/polym16030315

**Published:** 2024-01-24

**Authors:** Lili Chen, Yincai Wu, Yuntian Guo, Xiaofeng Yan, Wenliang Liu, Si Huang

**Affiliations:** 1Key Laboratory of Advanced Packaging Materials and Technology of Hunan Province, Hunan University of Technology, Zhuzhou 412007, China; lily_jasmine_001@163.com (L.C.); gyt.hyacinth@163.com (Y.G.); 13699429560@163.com (X.Y.); 2Art Institute, Hengyang Normal University, Hengyang 421010, China; 3Xiamen Institute of Rare Earth Materials, Chinese Academy of Sciences, Xiamen 361021, China; xmwuyincai@fjirsm.ac.cn

**Keywords:** navel orange peel pomace, soluble dietary fiber, nanocellulose, best formula

## Abstract

The packaging problem with petroleum-based synthetic polymers prompts the development of edible packaging films. The high value-added reuse of navel orange peel pomace, which is rich in bioactive compounds, merited more considerations. Herein, nanocellulose (ONCC) and soluble dietary fiber (OSDF) from navel orange peel pomace are firstly used to prepare dietary fiber-based edible packaging films using a simple physical blend method, and the impact of ONCC on the film’s properties is analyzed. Adopting three methods in a step-by-step approach to find the best formula for edible packaging films. The results show that dietary-fiber-based edible packaging films with 4 wt.% ONCC form a network structure, and their crystallinity, maximum pyrolysis temperature, and melting temperature are improved. What’s more, dietary-fiber-based edible packaging films have a wide range of potential uses in edible packaging.

## 1. Introduction

Due to their pollution-free, biodegradable, and edible qualities, edible packaging films had emerged as a major research hotspot in the field of food packaging in response to the growing resource waste and environmental pollution caused by petroleum-based synthetic polymer packaging [1,2]. With the inclusion of safe and edible plasticizers, thickeners, or cross-linking agents, as well as other small molecules, edible packaging films were prepared using natural biopolymers like edible proteins [3], lipids [4], polysaccharides, and their derivatives [5,6] as film-forming substrates. Following blending, ultrasound, drying, and other procedures, the interaction and cross-linking between the components that form the film were encouraged and followed drying, a film with specific mechanical properties, selective permeability, and dense network structure was formed [7,8]. The edible packaging film prepared using the solution casting method could be consumed along with food, offering a new form of food packaging and effectively achieving green packaging [9]. This film also had a lot of potential for use in ensuring food safety and promoting environmental friendliness.

With approximately 35% of banana waste, 25% of apple waste, 50% of citrus waste, and 40–45% of pomegranate waste, it had been thought in recent years that the waste produced by the fruit processing sector was one of the major issues generating worldwide resource waste and environmental damage [10,11]. With one-third of them being utilized for production and processing, navel oranges were one of the most significant fruit crops in the world, producing a lot of trash from peel pomace. With low utilization rates, the majority of the navel orange peel pomace was either processed as waste or used as animal feed and fertilizer, leading to considerable financial losses and environmental problems [12]. However, compared to other biomass such as cereals and vegetables, navel orange peel pomace included more bioactive substances in abundance, such as dietary fiber and polyphenols [13]. Among them, the total amount of dietary fiber in navel orange peel pomace reached 61.26 g/100 g, with a high soluble dietary fiber proportion of 19.56% [14].

Dietary fiber (DF) was a word that, since its introduction in 1953, had been used to describe plant cell wall components that had been altered to become plant polysaccharides that could not be absorbed and digested by the human gastrointestinal tract [15]. In 2000, the American Association of Grain Chemists (AACC) described DF as the edible component or comparable carbohydrate of a plant [16]. However, DF could not be broken down and absorbed in the human small intestine. Due to its water solubility, dietary fiber was separated into soluble dietary fiber (SDF) and insoluble dietary fiber (IDF). SDF (soluble dietary fiber) had high solubility in water, including components such as pectin, oligosaccharides, β-glucan, and inulin. It possessed gel-forming properties and good film-forming capabilities [17]. Reports indicated that pectin extracted from grapefruit peel exhibited superior thermal stability and physicochemical properties compared to commercially available citrus pectin, especially in terms of its methoxylation degree, anhydrous uronic acid, and moisture content [18]. IDF, such as cellulose, hemicellulose, lignin, and other types of IDF, were insoluble in water, and they could be prepared into nano cellulose [19] as a filler to improve the mechanical capabilities, barrier properties, and other characteristics of edible packaging films [20]. According to several studies [21,22], DF, which has stable physicochemical properties, serves as the “seventh largest nutrient” and has a variety of physiological-regulating, health-improving, and disease-prevention capabilities. In addition to increasing the amount of DF consumed, the use of DF as a raw material for preparing edible packaging films also made it a more generally available, high-performing, and inexpensive option.

In this study, navel orange soluble dietary fiber (OSDF) and navel orange insoluble dietary fiber (OIDF) extracted from navel orange peel pomace (OP) were used for the first time to prepare a green and biodegradable edible packaging film. This not only added nutritious value to food packaging but also offered a cutting-edge method for the sustainable use of fruit waste. The specific substrates utilized were OSDF and sodium alginate (SA), and the filler used was nanocellulose (ONCC), which was prepared from OIDF using an ultrasonic-aided organic acid process. Via simply physical blending, good-performance OSDF/SA/ONCC edible packaging film was prepared. Three methods were employed in a step-by-step process to find the optimal recipe for preparing OSDF/SA/ONCC edible packaging film. The microstructure, chemical content, and thermal properties of the OSDF/SA/ONCC edible packaging films were analyzed using SEM, TEM, XRD, and TGA techniques. The rheological properties of the edible packaging film solution were analyzed using a rheometer. The findings indicated that the OSDF/SA/ONCC, composed of dietary fiber and containing 4 wt.% ONCC, demonstrated exceptional performance. This study not only maximized the utilization of dietary fiber from navel orange peel pomace in edible packaging films, but it also offered a novel theoretical framework for making edible packaging films.

## 2. Experimental

### 2.1. Reagents

The navel orange peels were obtained from Guoran Food Co., Ltd. (Ganzhou, Jiangxi, China). The food-grade enzymes, comprising lipase (1.0 × 10^5^ U/g), papain (1.0 × 10^5^ U/g), mesophilic α-amylase (1.0 × 10^4^ U/g), and cellulase (1.1 × 10^4^ U/g), were sourced from Xiasheng Enzyme Biotech Company Limited in Cangzhou, Hebei, China. Additionally, oxalic acid, glycerol, and sodium alginate were obtained from Aladdin, situated in Shanghai, China.

### 2.2. Preparation of OSDF, OIDF and ONCC

By utilizing food-grade lipase, papain, medium-temperature amylase, and cellulase throughout the extraction process, OSDF and OIDF were extracted using an ultrasound-assisted composite enzyme method. Oxalic acid was utilized to prepare ONCC using OIDF as the raw material.

### 2.3. Preparation of the Edible Packaging Films

Use SA and OSDF as film-forming substrates and set the films’ dry weight. Prepare the film solution at a 1:25 (*w*/*v*) material-to-liquid ratio. Weigh 0.3 g of OSDF and SA, blended at room temperature for 30 min; then, after achieving a uniform mixture, add glycerol at concentrations of 5, 15, 25, 35, and 45 wt.% (based on the dry weight of the film). After stirring for 10 min, the OSDF/SA film solution was obtained; add a certain amount of ONCC suspension (1%, 2%, 3%, 4%, or 5% *w*/*w*) into the film solutions and stir for 45 min to obtain OSDF/SA/ONCC film solutions. Dry the film solutions separately in an air-drying oven at 40 °C for 12 h. After drying, the edible packaging film was placed in a constant temperature and humidity chamber for more than 24 h to test its various properties. A schematic diagram of the extraction and preparation process in this study is shown in Figure 1.

### 2.4. Calculation of the Overall Performance Score

Use principal component analysis and the membership function synthesis method to calculate the overall score of the edible packaging film performance index. The principal component analysis was used to determine the weights of the importance of each index. The tensile strength (TS), elongation at break (EB), oxygen transmission rate (OR), light transmission rate (LR), and heat-seal properties (HP) of food packaging films were made for comprehensive evaluation using the comprehensive scoring method of membership degree (P). The TS, EB, LR, and HP should be as large as possible, the membership degree was calculated using Equation (1); the lower the OR index, the higher the membership degree. The membership degree was calculated using Equation (2). On this basis, the comprehensive performance score S of the edible packaging film was further calculated (Equation (3)).
(1)$$P=\;\frac{A_{i}-A_{min}}{A_{max}-A_{min}}$$
(2)$$P=\frac{A_{max}-A_{i}}{A_{max}-A_{min}}$$
S = a_1_P_1_ + a_2_P_2_ + a_3_P_3_ + a_4_P_4_ + a_5_P_5_(3)
In the formula, *A_i_* represents the index value, and *A_min_* and *A_max_* represent the minimum and maximum values of the same index, respectively; a_1_, a_2_, a_3_, a_4_, and a_5_ represent the weights of TS, EB, OR, LR, and HP, respectively; and P_1_, P_2_, P_3_, P_4_, and P_5_ represent the membership degrees of TS, EB, OR, LR, and HP, respectively.

### 2.5. Response Surface Optimization Design

Based on the results of single-factor experiments, according to the Box-Behnken design principle, using the OSDF addition amount (A), ONCC mass fraction (B), and Gly mass fraction (C) as independent variables, and the comprehensive score S of the performance of the edible packaging film as the response value, a three-factor, three-level response surface experiment was designed. In Table 1, the factor level design was shown.

### 2.6. Characterization

#### 2.6.1. Microstructure

To determine the particle size and Zeta potential of the ONCC suspension, a Zetasizer Nano ZS90 nanoparticle size potentiometer, based in Malvern, Worcestershire, UK, was used. The microscopic morphology of OIDF, ONCC, and edible packaging films was analyzed using transmission electron microscopy (Hitachi H-7650, Tokyo, Japan) and field emission scanning electron microscopy (Apreo S, Waltham, MA, USA).

#### 2.6.2. Chemical Structure

A Fourier transform infrared spectrometer, the Nicolet 380, located in Madison, WI, USA, was utilized for capturing the FTIR spectrum of the samples within the scanning range of 4000 to 400 cm^−1^. The samples’ crystallinity was analyzed using an X-ray diffractometer (Ultima IV, Akishima, Tokyo, Japan); then, Segal’s empirical equation (Equation (S1)) was used to calculate the crystallinity index (CrI).

#### 2.6.3. Thermal Properties

Thermogravimetric analysis (TGA) and differential scanning calorimetry (DSC) were carried out on the thermogravimetric analyzer (Q50, New Castle, DE, USA) and DSC analyzer (Q20, New Castle, DE, USA) at a heating rate of 10 °C/min while nitrogen was present.

#### 2.6.4. Rheological Properties

The rheological properties of the film solution were analyzed using a rheometer (ARES-G2, New Castle, DE, USA) and a cone plate (diameter 25 mm, gap 0.052 mm). A temperature of 25 °C and shear rates ranging from 0.01 to 100 s^−1^ were used to analyze the steady-state rheological characteristics of the film solution. The static rheological curve of the film liquid was fitted using the Ostwald-de Wale power law equation (Equation (4)), with viscosity represented by either the apparent viscosity $\eta_{a}\left(\dot{\gamma }\right)$ (Equation (S2)) or the differential viscosity $\eta \left(\dot{\gamma }\right)$ (Equation (S3)).
(4)$$\sigma =K{\dot{\gamma }}^{n}\;\mathrm{or}\;\eta_{a}\left(\dot{\gamma }\right)=K{\dot{\gamma }}^{n-1}$$
In the formula, σ, $\dot{\gamma }$, *K*, and *n*, respectively, represent shear stress, shear rate, consistency index, and non-Newtonian index.

The Cox-Merz rule stated that when the $\dot{\gamma }$ and oscillation frequency ω were comparable, the steady-state shear viscosity $\eta \left(\dot{\gamma }\right)$ was approximately equal to the absolute value of the complex viscosity |η(ω)|, as shown in Equation (5).
(5)$$\eta \left(\dot{\gamma }\right)={\left|\eta \left(\omega \right)\right|}_{\left(\omega =\dot{\gamma }\right)}={\sqrt{{\left(\frac{G^{\prime }}{w}\right)}^{2}+{\left(\frac{G^{\prime \prime }}{w}\right)}^{2}}}_{\omega =\dot{\gamma }}$$
In the formula, G′ and G″, respectively, represent the storage modulus and the loss modulus.

### 2.7. Data Analysis

The analysis of the experimental data was conducted using the IBM SPSS Statistical 26 software, based in Armonk, NY, USA, with one-way ANOVA employed to assess the statistical significance of differences in the data. For each experiment, a minimum of three repetitions was carried out, and the outcomes were expressed as the average value plus or minus the standard deviation.

## 3. Results and Discussion

### 3.1. Calculation Results and Analysis of Comprehensive Performance Score

Use Equations (1) and (2) to calculate the membership degree of each performance index based on the single-factor experimental data. Principal component analysis was carried out on edible packaging films using software IBM SPSS Statistic 26 after extracting six sets of data from single-factor experimental data (Appendix A). To get the correlation matrix (Appendix A), feature roots, and contribution rates of the principal components (Table 2), the five performance indexes of edible packaging films were transformed and dimensionally reduced. Two main components were found after analysis. Through analysis, the two principal components TS and EB could be extracted, each with an eigenvalue exceeding 1. Together, they had a cumulative variance contribution rate of 82.316%, suggesting that TS and EB were capable of reflecting the information of all performance indicators of edible packaging films [23]. The characteristic roots, variance contribution rates, and load matrix coefficients (Appendix A) of each principal component could be utilized for calculating the weights of the five performance indicators of edible packaging films (Appendix A). Equations (3) and (6) could be integrated to determine the S.
S = 0.307P_1_ + 0.195P_2_ + 0.286P_3_ + 0.021P_4_ + 0.195P_5_(6)

### 3.2. Response Surface Optimization Results and Analysis

Based on the Box Behnken central combination design concept, a 3-factor, 3-level experiment was created to better study the impact of the interaction between variables. Table 3 showed the experimental setup and findings. To obtain a quadratic polynomial regression model, perform multiple regression fitting using software Design Expert 10 on the test data in Appendix A.
S = 0.828 − 0.0775A + 0.005B + 0.0025C − 0.0025AB − 0.0025AC + 0.0175BC − 0.09775A^2^ − 0.15775B^2^ − 0.13275C^2^(7)

Analyze the variation and perform a significance test on the overall response performance score. Table 4 and Appendix A showed the findings of the aforesaid model’s fitting analysis, while Appendix A showed the residual distribution and fluctuation effect. The regression model had a *p* < 0.001 value, showing a significant difference in the model, as demonstrated in Table 4 and Appendix A. The mismatch term *p* = 0.3099 (i.e., *p* > 0.05) showed that the mismatch was not significant. The order of each factor’s influence on the overall performance score was A > B > C, as could be observed from the F-value. The model of R^2^ and R^2^_adj_ did not significantly differ, which showed that it had a good level of prediction accuracy. From Appendix A, it was observed that the actual values of S were evenly distributed, showing the model’s viability. As a result, the aforementioned model had a strong fit with actual experiments and could accurately analyze and forecast the overall performance score.

Create a three-dimensional spatial response surface map and contoured map using the response value performance overall score, OSDF addition amount, ONCC quality score, and Gly quality score. Based on the regression equation, predicted the effects of various factors on the performance overall score. Figure 2 shows that the trend of overall performance scores under different factors was essentially consistent, i.e., there was an interaction between the factors and the performance overall scores that showed a trend of first increasing and then decreasing with A, B, and C. The highest point on the graph depicted the maximum value within the selected range. The A was between 0.15 g and 0.18 g, the B was between 3.7% and 4.2%, and the C was between 22% and 27%, which resulted in a high-performance overall score, and the ability to obtain optimum process parameters. This experiment used the software Design Expert 10 to carry out extensive optimization using performance overall classification as the optimization aim. Appendix A showed the results of the optimization. When A was set at 0.17 g, B at 4%, and C at 25%, the model’s predicted total performance score amounted to 0.843. Based on the optimization results of the previously mentioned factors, validation experiments were conducted, and the results showed that the overall performance score was 0.812, 0.851, and 0.806, with an average overall performance score of 0.826, which was close to the model’s projected value. This shows that the response surface model-based method of complete performance score analysis and optimization is efficient and practical.

### 3.3. Micromorphological Analysis

Figure 3 showed the outcomes of the SEM and TEM analysis of the microstructure of OP, OIDF, and ONCC. The body of OP was sizable and had a wrinkled, curved surface (Figure 3a). The total dietary fiber content of OP (63.24 g/100 g) [24] was significantly higher than that of mango peel pomace (35.51 g/100 g) and watermelon pomace (47.48 g/100 g) [25]. And, it was also rich in OSDF (13.62 g/100 g) and OIDF (49.53 g/100 g). This suggested that, as a renewable cellulose resource, OP offered the potential for value-added usage. The OP was treated using an ultrasonic-assisted compound enzyme method, which led to the production of a rough and loose OIDF (Figure 3b), mainly as a result of the fundamental removal of lignin, hemicellulose, and fat from the OP [26]. ONCC, obtained from OIDF through oxalic acid hydrolysis, exhibited a needle-like structure, with lengths ranging from 100 to 1000 nm and a diameter of about 4 nm, as shown in Figure 3c. This uneven size, adhesion, and the stacking phenomenon were caused by the OIDFs’ tiny diameter and large specific surface area, as well as the crystalline and amorphous regions’ erratic sizes and the effect of van der Waals forces. Figure 3d showed the results of further research on the ONCC’s particle size and Zeta potential. The average particle size of ONCC, which was 308.20 nm, was concentrated between 100 and 1000 nm, which was comparable to the statistical findings of transmission electron microscopy. An essential metric for assessing the stability of the dispersion system was the zeta potential. The stronger the repulsion between charged particles and the higher the stability of the dispersion system, the larger the absolute value of the value. The Zeta potential of ONCC was −42.20 mV, which was close to the conclusion that the Zeta potential of nano-cellulose dispersion liquid was between −20 mV and −50 mV in related studied [27]. According to earlier research [28], dispersed particles were more likely to aggregate when the absolute value of the Zeta potential was less than 15 mV, while the dispersion system was very stable when the absolute value was greater than 30 mV. This shows that ONCC prepared with oxalic acid can be diffused steadily in deionized water.

To study the dispersion of ONCC in the OSDF matrix and the compatibility between OSDF and ONCC, SEM was used to characterize the surface and cross-sectional morphology of the edible packaging films OSDF/SA and OSDF/SA/ONCC. Figure 3e–f showed that, while the OSDF/SA edible packaging film had a uniform and smooth surface and a 34 μm thickness, the surface did exhibit some small holes and cracks. The OSDF/SA/ONCC edible packaging film’s surface structure became smoother and denser after the addition of ONCC, and its thickness rose to 65 μm (Figure 3g–h). This was because the void space in edible packaging films was effectively filled by ONCC small particle molecules. The small amount of white granular materials that were discovered on the OSDF/SA/ONCC edible packaging film may have been brought on by the agglomeration of a few ONCC. Additionally, the addition of ONCC did not result in any obvious voids in the edible packaging film, indicating that ONCC was well-distributed within the OSDF matrix and has good biocompatibility with OSDF. The composite membrane made from nano-cellulose extracted from pineapple peel pomace and gelled adhesive also shows a similar effect [29].

### 3.4. Chemical Structure Analysis

Figure 4a showed the findings of an investigation into the interactions between the elements of edible packaging film. The ONCC showed the typical cellulose I absorption bands at 3350 cm^−1^, 2905 cm^−1^, and 1637 cm^−1^ [30]. The absorption maxima at 1430 cm^−1^ and 1060 cm^−1^, respectively, were caused by the vibrational absorption of the C-H bond bending and alcohol hydroxyl C-O stretching in cellulose molecules. At 895 cm^−1^, they were brought on by the hetero-cephalic vibration of the glycosidic bond group, which corresponds to the C-H bond deformation vibration in cellulose [31]. The OH...O stretching vibration of the O-H group and hydrogen bond, as well as the symmetric stretching vibration absorption peak of the C-H bond in methylene, were the absorption peaks of the OSDF/SA edible packaging film at 3283 cm^−1^ and 2924 cm^−1^ [32], respectively. The asymmetric and symmetric stretching vibrations of the carboxylate group were responsible for the absorption peaks at 1604 cm^−1^ and 1412 cm^−1^ [33], respectively. The C-O-C stretching vibration was responsible for the absorption peak at 1024 cm^−1^, whereas the C-H bond’s bending vibration was responsible for the absorption peak at 855 cm^−1^. There were no new peaks in the FTIR spectrum of the OSDF/SA/ONCC edible film when compared to the FTIR spectrum of the OSDF/SA edible packaging film, showing that there was no chemical interaction during the creation of the film between OSDF, SA, and ONCC. The ONCC contained a lot of hydroxyl groups that caused the O-H group peak strength in the edible packaging film of OSDF/SA/ONCC to increase, suggesting that covalent or hydrogen bonds were forming between the molecules of the edible film [34]. However, no new absorption peak was produced, further demonstrating the compatibility of ONCC and OSDF. 

The results of an XRD examination of the crystal structures of the ONCC, OSDF/SA, and OSDF/SA/ONCC edible packaging films were shown in Figure 4b. ONCC was composed of many cellulose polymorphisms. The usual high crystallinity cellulose type I structures were represented by the characteristic diffraction peaks at 2θ = 15.28°, 22.18°, and 34.12°, which correspond to the (101), (002), and (040) crystal planes, respectively [35]. The crystallinity index (CrI) of ONCC was estimated to be 20.87% using Segal’s empirical calculation. The edible packaging films OSDF/SA and OSDF/SA/ONCC both showed broad diffraction peaks at 2θ = 20.36°, indicating that they were in an amorphous phase [36] and that the interaction between OSDF and ONCC did not alter ONCC’s crystal structure, which was consistent with Zhang et al.’s study findings [37]. The main diffraction angle of the OSDF/SA/ONCC edible packaging film changed from 20.36° to 19.96° in comparison to OSDF/SA edible packaging film, and the intensity slightly increased, showing an increase in the crystallinity index (31.23%). This result indicates that incorporating ONCC enhances the crystalline structure of edible packaging films, and that augmenting crystallinity plays a crucial role in improving the mechanical strength of the composite films [38]. Additionally, OSDF/SA and OSDF/SA/ONCC edible packaging films’ half peak widths were 0.177 and 0.157, respectively, showing that the OSDF/SA/ONCC edible packaging films’ crystal structure was more ideal.

### 3.5. Thermal Properties Analysis

Figure 4c showed the TG and DTG curves for the edible packaging films ONCC, OSDF/SA, and OSDF/SA/ONCC. Within a temperature range of 30 °C to 600 °C, all samples showed multi-stage thermal degradation behavior. Under 100 °C, all samples’ weights gradually decline as the temperature rises, losing 1.47%, 5.91%, and 6.38% of their initial weights, respectively. Water evaporation and the dissolution of intermolecular hydrogen bonds were the causes of this occurrence [39]. The DTG curve showed distinctive peaks at temperatures between 100 °C and 500 °C, the TG curve steepened, and all samples went through macromolecular depolymerization and disintegration at this point. All samples did not alter in temperature when they reached 500 °C. They had residual rates that were greater than ONCC’s, which was 18.89%, including OSDF/SA and OSDF/SA/ONCC edible packaging films, at 26.79% and 25.78%, respectively. Additionally, OSDF/SA and OSDF/SA/ONCC edible packaging films showed weight losses of 33.81% and 37.66%, respectively, and maximum pyrolysis temperatures of 223.60 and 228.15 °C. ONCC showed a weight loss of 54.80% and a maximum pyrolysis temperature of 353.53 °C. Due to intermolecular hydrogen bonding and the presence of crystalline cellulose in ONCC, these results show that the addition of ONCC enhanced the thermal stability of edible packaging films [40].

The key to characterizing the processing temperature range and applications of composite films was the analysis of their thermal performance. DSC was used to examine the thermal behavior of the edible packaging films OSDF/SA and OSDF/SA/ONCC, and the findings were displayed in Figure 4d. The melting peaks and points of each sample were all visible; however, no crystallization peaks were found. It might take longer for the amorphous zone to rearrange due to the crystal zone’s shift from ordered to disordered after melting and the disturbance of the regular structure. In the temperature range of 120 to 140 °C, all samples exhibited wide endothermic peaks. ONCC had a melting temperature (T_m_) of 118.62 °C and a melting enthalpy of 148.17 J/g (ΔH_m_), whereas OSDF/SA and OSDF/SA/ONCC edible packaging films had T_m_ values and ΔH_m_ of 139.93 °C, 144.85 °C, and 180.95 J/g, 172.48 J/g, respectively. It was important to note that OSDF/SA/ONCC had the highest Tm value, showing that it had a complete internal crystal structure and higher mechanical characteristics, which corresponded with XRD data. According to our experimental findings, adding ONCC to the OSDF matrix could improve the thermal performance of edible packaging films by raising their T_m_ value and lowering their ΔH_m_. This might be because adding ONCC-made OSDF molecules had less motion capability [41].

### 3.6. Rheological Properties Analysis

The relationship between the mechanical properties and structure of biopolymer solutions might be seen in the rheological characteristics of film-forming solutions, which were crucial for optimizing film-forming processes. Figure 5a showed the correlation between the apparent viscosity and shear rate for the edible packaging film solutions OSDF/SA and OSDF/SA/ONCC. The viscosity of OSDF/SA edible packaging film solution showed a tendency of first leveling off and then declining, as could be seen. When the shear rate was between 0.1 and 10 s^−1^, the viscosity of the OSDF/SA edible packaging film solution was nearly constant, showing that the film solution was Newtonian; however, when the shear rate was between 10 and 100 s^−1^, the film solution entered the shear thinning region, where the molecular entanglement effect was evident and had significant non-Newtonian characteristics. After adding ONCC, the viscosity of the film solution quickly decreased as the shear rate increased, and its non-Newtonian features became more significant. In film-forming solutions based on polysaccharides, similar results had been found [42]. The film solution changed after the addition of ONCC into a suspension system and formed new hydrogen bonds, which increased viscosity [43]. However, when subjected to shear forces, the network structure of the film solution was disrupted, causing ONCC to arrange in an orderly manner, reducing resistance, and consequently leading to a decrease in viscosity [44]. The OSDF/SA/ONCC edible packaging film solution’s n value remained constant over the shear rate range, and the Ostwald-de Wale power law equation was utilized for fitting, with *n* = 0.12, k = 2.65, and apparent viscosity $\eta_{a}\left(\dot{\gamma }\right)=2.65\;{\dot{\gamma }}^{-0.88}$. The OSDF/SA edible packaging film solution’s rheological characteristics could be separated into two stages, with a critical shear rate of roughly 10 s^−1^ and *n* values of 1 and 0.72, respectively.

It was possible to find the linear viscoelastic region of the edible packaging film solution using amplitude scanning. The edible packaging film solution for OSDF/SA was shown in Figure 5b to be in a linear region across the whole testing range. After adding ONCC, the linear range of the film solution was between 0.1% and 1%.Therefore, the OSDF/SA edible packaging film solution was chosen with a final strain value of 3% and the OSDF/SA/ONCC edible packaging film solution with a frequency scanning strain value of 1%. Dynamic frequency scanning enables the revelation of interactions and structural features among various constituents [45]. Figure 5c showed that in the absence of ONCC in the film fluid its G′ was less than G″, indicating a liquid state. The G′ and G″ of the OSDF/SA/ONCC edible packaging film solution both increased with the frequency. Since G″ was less than G′ at low frequencies, it indicated that the film solution exhibited solid-like properties [46]. This might be because the addition of ONCC caused the gel network structure of the film liquid to form. The microstructure of the film solution and the mechanical properties of edible packaging films were improved by the addition of ONCC, which increased the G′ and G″ of the film solution, showing that ONCC was uniformly dispersed in the matrix and formed a significant number of hydrogen bond network structures with OSDF [47]. The Cox-Merz rule did not apply to the edible packaging film solution, as shown in Figure 5d, where it was clear that the complex viscosity and apparent viscosity of the film solution did not overlap. In the absence of ONCC in the film fluid, its phase angle δ approaches 90°, but the δ of the OSDF/SA/ONCC edible packaging film solution dramatically fell at the low modulus end, as seen by the δ–G^*^ relationship curve (Figure 5e). OSDF/SA/ONCC had solid-like qualities as a result of the insertion of filler particles [48]. The OSDF/SA edible packaging film solution showed liquid characteristics and steady-state viscosity, as shown by the η^*^–G^*^ relationship curve (Figure 5f). After adding ONCC, the film fluid would start to flow or deform under certain conditions, no longer maintaining its original form [49].

## 4. Conclusions

In this work, an edible packaging film using OSDF as the substrate and ONCC as the reinforcing material was successfully prepared for the first time. The edible packaging film containing 4 wt.% ONCC was determined to have the highest overall performance score. The crystallinity index, maximum pyrolysis temperature, and melting temperature of edible packaging film all increased after the addition of ONCC by 2.51%, 4.55 °C, and 4.92 °C, respectively, and the phenomena of film solution shear thinning became more apparent. The above discussion suggests that using dietary fiber from orange peel pomace as an edible packaging material represents a significant advancement in the development of sustainable materials for food packaging.

## Figures and Tables

**Figure 1 polymers-16-00315-f001:**
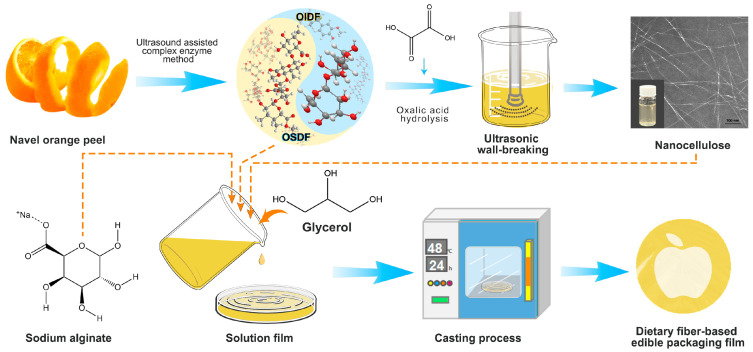
A schematic diagram of the extraction and preparation process in this study.

**Figure 2 polymers-16-00315-f002:**
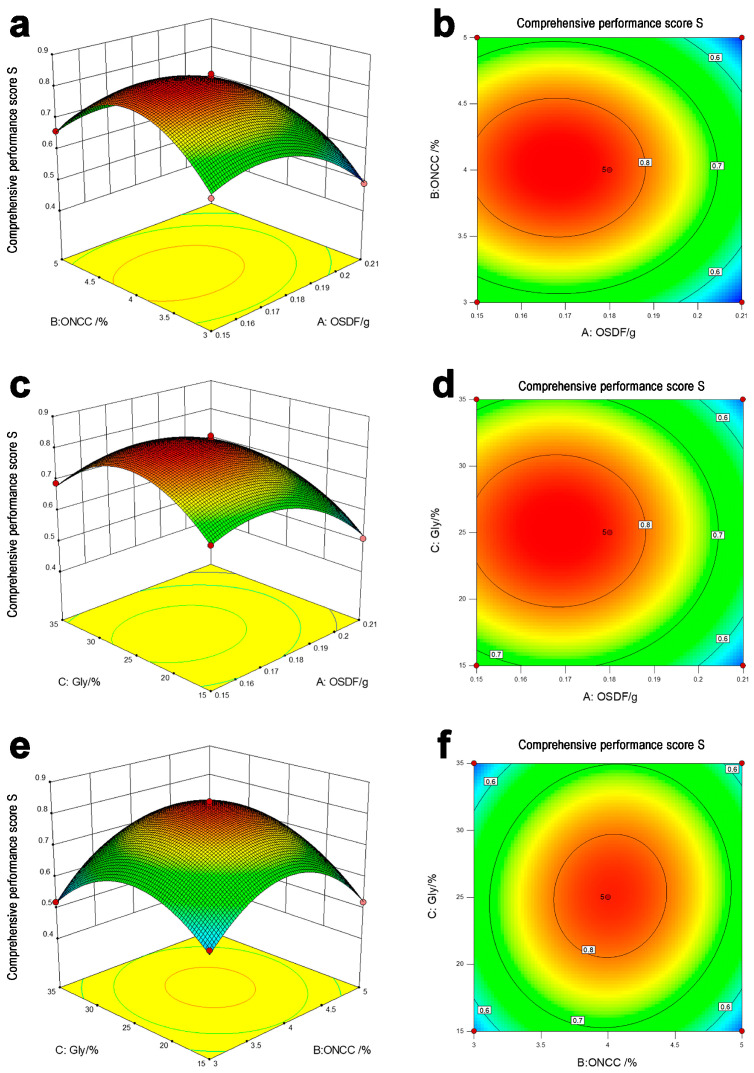
(**a**,**b**) The influence of the interaction between A and B on S. (**c**,**d**) The influence of the interaction between A and C on S. (**e**,**f**) The influence of the interaction between B and C on S.

**Figure 3 polymers-16-00315-f003:**
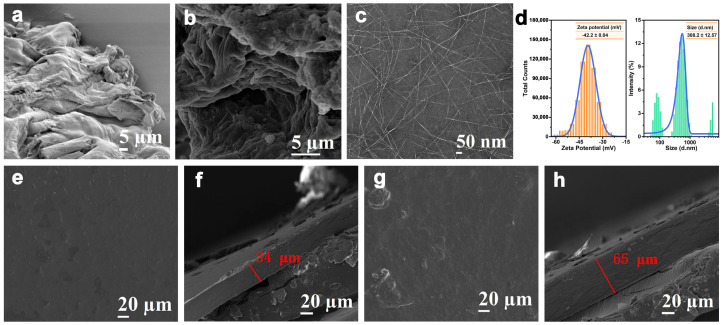
SEM micrographs of OP (**a**) and OIDF (**b**); (**c**) TEM image; (**d**) Particle size distribution and Zeta potential of ONCC; Surface (**e**) and cross-section (**f**) SEM images of OSDF/SA; Surface (**g**) and cross-section (**h**) SEM images of OSDF/SA/GNCC.

**Figure 4 polymers-16-00315-f004:**
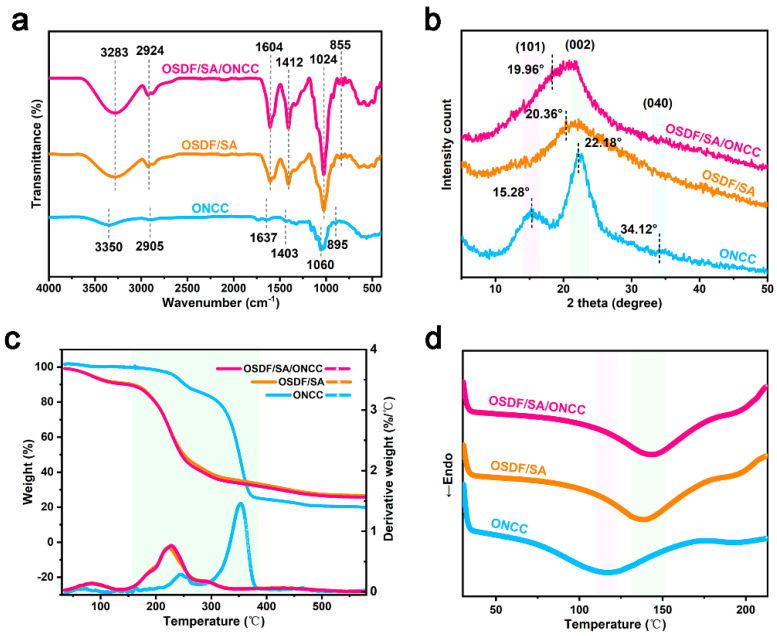
(**a**) FTIR spectra; (**b**) XRD; (**c**) TGA; (**d**) DSC.

**Figure 5 polymers-16-00315-f005:**
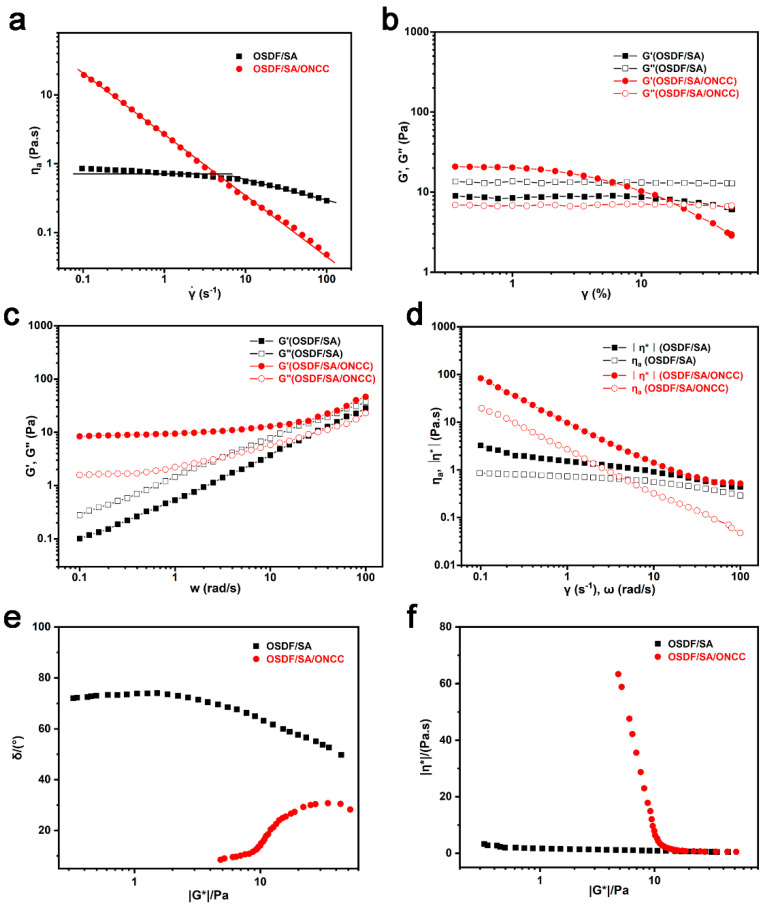
(**a**) Apparent viscosity−shear rate relationship curve; (**b**) G′/G″—oscillation strain relationship curve; (**c**) G′/G′—angular frequency relationship curve; (**d**) Cox-Merz rule; (**e**) δ−G^*^ relationship curve; (**f**) η^*^−G^*^ relationship curve of film fluid.

**Table 1 polymers-16-00315-t001:** Factor and its level of response surface analysis.

Levels	OSDF/(g)	ONCC/(%, *w*/*w*)	Gly/(%, *w*/*w*)
−1	0.15	3	15
0	0.18	4	25
1	0.21	5	35

**Table 2 polymers-16-00315-t002:** Total variance explanation of the edible packaging film.

Component	Eigenvalue	Percentage of Variance/(%)	Cumulative/(%)
1	2.699	53.978	53.978
2	1.417	28.338	82.316
3	0.823	16.461	98.777
4	0.057	1.149	99.926
5	0.004	0.074	100.000

**Table 3 polymers-16-00315-t003:** Experimental design and results using the response surface methodology.

Trial Number	OSDF A/g	ONCC B/%	Gly C/%	Overall Performance Score S
1	1	0	−1	0.51
2	0	0	0	0.83
3	−1	0	−1	0.67
4	1	1	0	0.51
5	0	1	−1	0.52
6	0	−1	−1	0.56
7	−1	−1	0	0.63
8	0	−1	1	0.52
9	0	0	0	0.84
10	0	0	0	0.81
11	0	0	0	0.84
12	1	0	1	0.52
13	0	0	0	0.82
14	−1	1	0	0.66
15	−1	0	1	0.69
16	0	1	1	0.55
17	1	−1	0	0.49

**Table 4 polymers-16-00315-t004:** ANOVA for response surface quadratic model.

Source	Sum of Squares	df	Mean Square	F Value	*p*-Value	Significance
Model	0.29	9	0.033	149.10	<0.0001	**
A	0.048	1	0.048	219.84	<0.0001	**
B	2.0 × 10^−4^	1	2.0 × 10^−4^	0.92	0.3706	
C	5.0 × 10^−5^	1	5.0 × 10^−5^	0.23	0.6470	
AB	2.5 × 10^−5^	1	2.5 × 10^−5^	0.11	0.7451	
AC	2.5 × 10^−5^	1	2.5 × 10^−5^	0.11	0.7451	
BC	1.225 × 10^−3^	1	1.225 × 10^−3^	5.60	0.0498	*
A^2^	0.040	1	0.040	184.07	<0.0001	**
B^2^	0.10	1	0.10	479.38	<0.0001	**
C^2^	0.074	1	0.074	339.48	<0.0001	**
Residual	1.53 × 10^−3^	7	2.186 × 10^−4^			
Mismatch term	8.5 × 10^−4^	3	2.833 × 10^−4^	1.67	0.3099	
Pure errors	6.8 × 10^−4^	4	1.7 × 10^−4^			
Totals	0.29	16				

* *p* < 0.05 means significance, ** *p* < 0.01 means extremely significance.

## Data Availability

Data are contained within the article and Appendix A.

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
