# Peer review of "Preparation and Characterization of Soluble Dietary Fiber Edible Packaging Films Reinforced by Nanocellulose from Navel Orange Peel Pomace"

_polymers, 2024, doi:10.3390/polym16030315_

Round 1

Reviewer 1 Report

Comments and Suggestions for Authors

The authors have done a lot of work on their paper on preparation of edible packaging films from navel orange peel, however I find the paper to be lacking. Please see below my comments in key areas which will help improve the manuscript.

1. In order to claim novelty, the authors must do a more in depth review of the preexisting work on utilizing citrus fruits for making packaging films. For instance a cursory search reveals the following paper which makes films from citrus peels. 

https://doi.org/10.1016/j.foodhyd.2022.107961

The authors should atleast compare their work with other similar research.

2. Section 2.3: "Weighed 0.3 g of OSDF and SA, reacted at room temperature for 30 min, then added 5, 15," please clarify what the reaction is.

3. The introductory section mentions that the films are high performance, but I did not see any performance metric like tensile strength (TS), elongation at break (EAB), oxygen transmission rate (OP), light transmission rate (LT), and heat-seal properties (HS). These have been mentioned but i did not see any of these values in text. Furthermore performance can only be judged by comparison with commercially available materials frequently used for food packaging.

Comments on the Quality of English Language

na

Author Response

Please consult the following document.

Reviewer 2 Report

Comments and Suggestions for Authors

In page 3 above Fig. 1 I prefer instead of "at constant temperature with a temperature of 25+-1oC" to write only "at constant temperature 25+-1oC".

In 3.1 and 3.2 the first sentences seem to be the titles of paragraphs following them. So, if this is the case highlight them with bold or italic.

At many places in units the necessary parts are not put in superscript.

In 3.4 in "OH-O stretching" write rather OH...O as this indicates hydrogen bond.

There is some problem with Equation 1 and 2 as how the authors wrote them they can be valid only in case when differences Ai-Amin and Amax-Ai are equal.

Equation 5 should be carefully checked as if complex viscosity is used, in expression on the right side one of the members should be multiplied with the imaginary unit i. Otherwise values of G' and G'' can not be determined in my opinion only sum of their squares.

Comments on the Quality of English Language

English is not problematic basically but the authors should go through the entire text as there are some serious mistakes. For example in presentation of results frequently can "showed" be found instead of "shows". In sentence found in 3.6 "Showed that the film solution exhibited..." the subject is absent.

Author Response

Please consult the following document.

Round 2

Reviewer 1 Report

Comments and Suggestions for Authors

I have reviewed the manuscript and I am ok with publication.